# Design of a Gas Permeation and Pervaporation Membrane Model for an Open Source Process Simulation Tool

**DOI:** 10.3390/membranes12121186

**Published:** 2022-11-25

**Authors:** Kouessan Aziaba, Christian Jordan, Bahram Haddadi, Michael Harasek

**Affiliations:** Institute of Chemical, Environmental & Bioscience Engineering E166, Technische Universität Wien, 1060 Vienna, Austria

**Keywords:** simulation, membranes, membrane model, DWSIM, gas permeation, pervaporation

## Abstract

Gas permeation and pervaporation are technologies that emerged several decades ago. Even though they have discovered increasing popularity for industrial separation processes, they are not represented equally within process simulation tools except for commercial systems. The availability of such a numerical solution shall be extended due to the design of a membrane model with Visual Basic based on the solution-diffusion model. Although this works approach is presented for a specific process simulator application, the algorithm can generally be transferred to any other programming language and process simulation solver, which allows custom implementations or modeling. Furthermore, the modular design of the model enables its further development by operators through the integration of physical effects. A comparison with experimental data of gas permeation and pervaporation applications as well as other published simulation data delivers either good accordance with the results or negligible deviations of less than 1% from other data.

## 1. Introduction

Membranes are already widely considered an alternative separation route alongside conventional separation technologies, such as cryogenic and adsorptive technologies, delivering comparably sufficiently high purities for many subsequent applications. Particularly advantageous compared to the alternatives mentioned is the easy maintenance, facile operation, small size, and low energy consumption, while a separation of liquid–liquid, vapor–liquid, and also gaseous mixtures is possible [1,2,3,4].

Within the framework of the design of technical processes, process simulation (PS) takes over many relevant functions. PS is a comparably cost-efficient and decisive tool in a company with experimental and plant engineering data for the design, operation, and optimization fields of process engineering. It offers the opportunity to predict the behavior of single unit operations (UOs) and the results of full processes as long as the models are available. It provides the opportunity to apply sensitivity analyses (SA), multivariate optimization, and copes relatively rapidly with design-of-experiments compared to experimental work. In addition, didactic purposes can be pursued with the use of process simulation.

Nevertheless, drawbacks can arise from those with comparably weak capabilities limiting PS’s expressive power. However, some UOs qualities may increase their predictive accuracy through further development and validation with test cases. Furthermore, convergence problems, the possible necessity of preconditions, occurring errors, and limited access to several PS software and its UOs are already adverse but improvable aspects. Yet, the predictions obtained by PS are increasingly indispensable in the technical industry [5].

Since the demands of PS users can vary significantly depending on the process that shall be designed or optimized, there is also a variable selection of UOs each PS software offers. The focus of this work lies within the development of membranes for an open-source tool. Several development approaches for designing and using membrane models in process simulation tools have been discovered and explored. However, the proposed model combines object-oriented methods to directly introduce a numerical solution for multi-component and multi-stage membrane separation into an open-source process simulation tool, while laying the groundwork for further developments by providing the complete source code as Appendix A. This modularity offers extensive development opportunities for the simulation and membrane research community, which is considered a novel development.

### 1.1. State of the Art: Mathematic Solutions for Membrane Models

Even though several publications already report models and implementations of gas permeation of binary systems, there is far less information about simulations to separate multi-component systems with membranes. Even though some information about the modeling is available in the literature, it is still constrained [6,7,8,9,10].

Pan et al., delivered a comparably early approach for the mathematical modeling of gas permeation for the co-current- and counter-current flow configuration model based on the solution-diffusion model compared with experimental data from a field pilot-plant recovering Helium (He) from natural gas. Pan et al.’s work considered the residues concentrations as the parameter to be set, yielding the permeate pressure, composition, and required fiber length. Since many membrane manufacturers do not offer the opportunity to provide custom fiber lengths, the applicability of this specific result may be limited [11].

Kundu applied the solution-diffusion model to separate multi-component gaseous mixtures with polymeric membranes to calculate the enrichment of Methane (CH_4_) from biogas. A mathematical model is proposed using ordinary differential equations (ODE) for co-current- and counter-current flow configurations. The numerical solving technique, which uses Gear’s method, demands the feed composition and the ideal selectivity referenced to a single compound. It describes the flux, the permeates flow rate, the composition of both, the permeate and the retentate, and the pressure build-up within the fiber [12].

Rezakazemi et al., proposed a mathematical 2D model that was handled with CFD-solving technology based on the Navier-Stokes equations to model HFMC for natural gas sweetening. Validation with experimental data is described as in “good agreement” with the simulation results, indicating that the model could also be used as a predictive instance for process designs [13]. Farno et al., provided a 3D gas permeation PDMS membrane model derived from mass transport and momentum transfer equations, also considering the diffusivity for a ternary system. An artificial neural network’s contribution to improving the prediction of the permeates composition, permeabilities, and solubilities was also discussed. The results were in good agreement with experimental findings [14]. Haddadi et al., introduced a CFD algorithm for gas permeation simulations based on a multi-compartment approach. Each region can be regarded as its standalone operation inheriting its turbulence models, and thermophysical properties. The algorithm was implemented in the open-source CFD environment OpenFOAM [15].

Katoh et al., proposed the tank-in-series method combined with the Runge–Katta integration method to solve the governing ODEs to handle the dynamic and non-ideal mixing behavior of the species across the membrane [16]. Elshof et al., described a mathematical model for the separation of water and organic solvents with a microporous silica membrane based on the Maxwell-Stefan model. The model relies on Maxwell-Stefan diffusion coefficients, which describe the transport through the microporous barrier, also considering the friction during this phenomenon [17]. Makaruk et al., developed a numerical approach for the calculation of co-current- counter current and cross-flow gas permeation membranes, which were also verified with experimental results [18].

Finally, it remains to mention the availability of further models to describe mass transfer across a membrane depending on the membrane type, such as the pore-flow model, which is occasionally applied for liquid binary mixtures and porous membranes. [19]

The following list shall highlight some advantages and drawbacks of the aforementioned modeling solutions:Solutions-Diffusion modelUndoubtedly, the solution-diffusion model (SDM) is one of the most widely used and adapted models found in the literature and process simulators for gas permeation and pervaporation applications. The description of the product streams is usually possible depending on the feed concentration, permeance, and a set product value, such as the concentration of the permeate or the pressure difference. Theoretically, there is no limit to the number of compounds the model can handle. The limited accuracy of the model can be a drawback. However, its relative simplicity makes it more likely to be combined with customized solving routines. Most models to follow in Section 1.3 are based on further developments of the solution-diffusion model;Maxwell–Stefan TheoryAs a theory that also incorporates the migration in dependence of friction and influences of the interaction of components and membrane, the Maxwell-Stefan theory is comparably more complex than the SDM. Theoretically, the model can process multi-component gas permeation and pervaporation systems. An advantage of the Maxwell-Stefan theory is the opportunity to obtain comparably more accurate information about non-ideal multi-component systems from the results of single-component calculations and an estimation of the systems’ selectivities [20];Pore flow ModelThis model is usually introduced as a theory for porous membranes separating liquid mixtures. In comparison, it is based on the migration of permeating components through theoretical pores on the membrane’s surface through capillary. Several adaptions of the underlying mathematical relations are to be found in the literature, such as the equation of Ergun, Carman–Kozeny, or Darcy’s law. While it is considered to perform roughly equivalently accurately, it shares the identical drawback as the SDM in that it majorly considers the pressure and concentration gradient. However, depending on the mathematical background, it may give information about the pressure drop due to the geometry of the membrane itself [19,20,21];Object-oriented ProgrammingEven though modeling with object-oriented programming methods is not an actual calculation routine similar to the earlier mass transfer methods, it can be a powerful approach for modeling in general. While calculation routines and algorithms are the center of interest for developers, the product mainly displays the model itself and its handling. This may indicate an increased complexity since other routines must be implemented then the pure numeric solution. However, this also poses a huge potential since every method can communicate with each other [22].

### 1.2. State of the Art: Membrane Models in Process Simulators

Aspen Technology is a well-known provider of two proprietary process simulators, Aspen Plus and Aspen Hysys. The primary application fields are process design and optimization in chemical engineering industries [23]. Many PS tools have an interface for integrating custom models or licensed addons, as Aspen Technologies does. The Aspen Custom Modeler belongs to the Aspen software package and is a commonly used solution for integrating user-generated models [24,25]. Additional integration methods in the literature use Aspen technology products, such as FORTRAN or MATLAB [26].

Nonetheless, adverse effects are reported using Aspen Hysys in combination with MATLAB, Excel, Python, and C# [27]. According to custom modeling and membranes, several models have already emerged for Aspen Plus and Aspen Hysys. A flat sheet gas permeation membrane model was implemented into Aspen Plus with the Aspen custom modeler based on the fugacities and mixture diffusivities derived from Blanc’s law. The predictive performance for the residuals molar fraction is described to be accurate [28].

ChemBrane is a custom membrane model developed by Graigner in 2007 which operates on Hysys based on a fourth-order Runga–Kutta method for the calculation of the flux. It is used to predict the flux for perfectly mixed, co-current, and counter-current flow configurations based on the permeabilities of each compound [29]. Cavalcanti et al., described a pervaporation membrane model based on the SDM integrated with the Aspen Custom Modeler. It was applied to predict the purification of EtOH with a polyetherimide membrane [30]. A high-pressure hollow fiber membrane contactor for natural gas purification was modeled for the PS tool gPROMS in a counter-current flow configuration [31]. AVEVA’s Pro/II is an equation-oriented steady-state PS tool that already inherits a membrane model developed based on a membrane reactor derived from Bishop et al. [32].

An overview of additional available PS tools which allow the development and integration of membrane models is available in the literature [33]. All described tools have in common that only proprietary PS tools are available today. This is an adverse and limiting precondition for multiple reasons. The accessibility for industry, researchers, and teachers is limited due to partially costly products. Further, the models are commonly provided as black box models. This excludes the opportunity to re-construct the algorithm to evaluate if the preliminarily determined preconditions are suitable for the model. In combination with the lost opportunity to modify or extend the model, many models’ capabilities are limited in many ways. Therefore, the drive for the development of an initially facile but continuously extended open-source membrane model for an open-source software was given.

### 1.3. Mathematical Background of the Model

Currently, the solution-diffusion model is reportedly a popular approach to model membrane separation. It is suitable for various gas and liquid feeds in membrane permeation applications. [34] The idealized counter- and co-current flow patterns are illustrated in Figure 1.

The local flux through the membrane is commonly described as follows Equation (1).
(1)d(xn)=−Q×A×(xP−yp)

The molar flow, molar fraction, and permeance are abbreviated as n, x, and Q. The feed pressure P, permeate pressure p, A, and y as the surface area and the permeate composition of the operation also determine the flux. The subscripted variables F, p, and r represent the feed, permeate, and retentate side of the corresponding value. The membrane area A can be calculated from the number of fibers ϵ, the inner diameter of the fiber $d_{fiber}$, and the fiber length $l_{fiber}$ as described in Equation (2).
(2)$$A=\epsilon \times d_{fiber}\times \pi \times l_{fiber}$$

Considering the fact that process simulation tools will be obliged to set multiple preconditions, such as plug flow, perfect mixing on the permeate and the feed/retentate side Equation (1) shall be integrated and set in relation with its feed, which shall result in a relation between the flux and the stage-cut. The molar fractions x and y are to be chosen depending on the flow configuration.
(3)$$\theta =\frac{yn_{p}}{x_{F}n_{F}}=\frac{AQ\left(xP-yp\right)}{x_{F}n_{F}}$$

Since the SDM is already well documented and published, this work will solely cover its implementation into the PS tool DWSIM, including its modifications and preconditions. However, the presented background is also well applicable to other PS tools. Alongside the solution-diffusion model itself, the following list of preconditions were also set:Negligible axial and radial pressure variation;Perfect mixing in every compartment;Negligible radial concentration variationPure component permeances are independent of feed composition, temperature, and pressure variations;Negligible deformation of the hollow fiber under pressure.

Since not only the feed composition but also the permeate composition is decisive for the driving force, it must also be considered in the calculations. With the previously stated upper assumptions, the permeate composition $y_{i}$ can be calculated as follows.
(4)$$y_{i}n_{p}=AQ\left(x_{r}P-y_{i}p\right)$$

For multi-component fluid mixtures, the permeate molar fraction for counter-current flow is calculated by forming the ratio of the components’ fluxes over the total flux [10]. The subscripts i, j, and k represent the ith, jth compound from a total number k of compounds.
(5)$$\frac{y_{i}}{1-y_{i}}=\frac{Q_{i}\left(x_{r,i}P-y_{i}p\right)}{\sum_{j\neq i}^{k}Q_{j}\left\{\left(1-x_{r,j}\right)P-\left(1-y_{j}\right)p\right\}}$$

A distinction must be made on whether a counter-current or co-current membrane shall be modeled. The driving force of the membrane model configured as a counter-current model competes with the permeate side aside from the feed’s end, which is indicated in Equation (6), which is a rearrangement from Equation (5).
(6)$$y_{i}=\frac{Q_{i}x_{r,i}\alpha_{i}\sum_{j\neq i}^{k}y_{j}}{\sum_{i\neq j}^{k}Q_{j}\left(x_{r,i}\alpha_{i}-y_{j}\right)+Q_{i}\sum_{j\neq i}^{k}y_{j}}$$

The driving force for the co-current flow mode is slightly different since it is the retentate end of the feed that corresponds with the actual permeate, as indicated in Equation (6).
(7)$$y_{i}=\frac{Q_{i}x_{F,i}\alpha_{i}\sum_{j\neq i}^{k}y_{j}}{\sum_{i\neq j}^{k}Q_{j}\left(x_{F,i}\alpha_{i}-y_{j}\right)+Q_{i}\sum_{j\neq i}^{k}y_{j}}$$

## 2. Materials and Methods

This section intends to provide an overview of the applied routines and methods to design and integrate the mathematical model into open-source PS software. Specific software was chosen to proceed with the approach, yet the presented procedures are applicable for any PS tool that can proceed or include numerical solvers.

### 2.1. Machine, Operating System and Software

Every applied software was operated on both a AsRock DeskMini A300 with a Ryzen 3 3400G and 32 GB of Ram and a Notebook Dell Inspiron 7415 with a Ryzen 5 5500U processor and 16 GB of RAM. All systems were operated with Windows 10. DWSIM was operated with version 8.0.3 on both systems. The Visual Basic(VB) code was scripted in Visual Studio 2017, its community distribution.

### 2.2. DWSIM

DWSIM is an open-source process simulation tool available on multiple platforms that was developed by Daniel Wagner Oliveira de Medeiros. It can be considered as a competitive alternative to other process simulators, such as Aspen Plus or PRO-II. It also operates its calculations as steady-state simulations. Since Version 6, dynamic simulations were introduced into DWSIM. With Version 7 DWSIM Pro was introduced, which offers various extensions, such as further Unit Operations (UO) and property packages. DWSIM provides the possibility to modify individual units as well as entire calculation routines of the flowsheet. Further, it already supports the opportunity to manipulate the communication between UOs and the flowsheet, and between UOs. The creation of whole UOs as custom plugins is an additional strength DWSIM offers. Yet, they must have been written in DWSIM’s native coding language, Visual Basic and C# [35].

### 2.3. Steps to Follow

#### 2.3.1. Work Environment

DWSIM was obtained from SourceForge [36]. The setup inherits Microsoft WebView 2 Runtime, ChemSep, CAPE-OPEN Type Libraries, Register Type Libraries, and DWSIM itself [35]. The initial step to set up the actual work environment is creating a VB Class project by configuring the project’s name, location and aiming. NET Framework. Targeting the same. NET Framework as the DWSIM is running is recommended. A selection of DWSIM-related dynamic link library files (.dll) will offer the opportunity to run DWSIM-associated functions within VB if implemented in the reference section. Since not every library is needed for the developed functions, the complete list of DLL’s is provided in Table A1.

#### 2.3.2. Implementation of General Models Related Functions

Since DWSIM is an open-source tool, there are also open libraries that can be applied to various models. Functions and code structures, such as the opportunity to process several properties of incoming materials streams or even the cosmetic design of the model itself, are provided to a fair extent. Table A2 contains the applied libraries.

In order to ultimately start the code, a couple of initial properties must be defined. Since the model of interest is a UO, it must also inherit the data structure of it, while it should also include the opportunity to apply the typical functions of other UOs in DWSIM. The two following lines of code can realize these:*Inherits UnitOperations.UnitOpBaseClass**Implements DWSIM.Interfaces.IExternalUnitOperation*

Following this, the first set of properties shall be declared here, such as the UO’s name and description and the category to which the UO belongs. Whenever a UO is initialized, it also inherits default values for its parameters, which can also be provided as early as this step.


*Public m_ResStageCut As Dictionary(Of String, Double)*

*Private Property UOName As String = “Membrane”*

*Private Property UODEscription As String = “Membrane Unit Operation”*

*Private _components As New List(Of String)*

*Private _componentsids As New List(Of String)*

*Private _components_re As New List(Of String)*

*Dim N0 As New Dictionary(Of String, Double)*

*Dim Colums, Rows As Integer*

*Public Overrides Property ComponentName As String = UOName*

*Public Overrides Property ComponentDescription As String = UODEscription*

*Public Overrides Property ObjectClass As SimulationObjectClass = SimulationObjectClass.Separators*

*Public Property Permeances As New Dictionary(Of String, Double)*

*Public Property AllPermeances As New Dictionary(Of String, Double)*

*Public Property PermeatePressure As Double = 100000.0*

*Public Property NumberFibers As Double = 100.0*

*Public Property InnerDiameterFibers As Double = 0.01*

*Public Property FiberLength As Double = 0.1*

*Public Property Chambers As Double = 1*

*Public Property StageCut As Double = 0.2*


One primary motivation for creating this UO was the opportunity to design the property editor (PE) in a fully customized way. Therefore, the PE will take its own descriptive part within the Appendix A. However, the PE also has to be linked to the main UO model, which usually shall also occur in the initial part of the code.


*Public Overrides Sub DisplayEditForm()*

*If editwindow Is Nothing Then*

*editwindow = New Editor() With {.HObject = Me}*

*editwindow.ShowHint = GlobalSettings.Settings.DefaultEditFormLocation*

*editwindow.Tag = “ObjectEditor”*

*Me.FlowSheet.DisplayForm(editwindow)*

*Else*

*If editwindow.IsDisposed Then*

*editwindow = New Editor() With {.HObject = Me}*

*editwindow.ShowHint = GlobalSettings.Settings.DefaultEditFormLocation*

*editwindow.Tag = “ObjectEditor”*

*Me.FlowSheet.DisplayForm(editwindow)*

*Else*

*editwindow.Activate()*

*End If*

*End If*

*FlowSheet.DisplayForm(editwindow)*

*End Sub*


On a regular application of the UO on a flowsheet, the PE must be frequently closed and opened, resulting in the necessity of implementing these functions.


*Public Overrides Sub UpdateEditForm()*

*If editwindow IsNot Nothing Then*

*If editwindow.InvokeRequired Then*

*editwindow.Invoke(Sub()*

*editwindow?.UpdateInfo()*

*End Sub)*

*Else*

*editwindow?.UpdateInfo()*

*End If*

*End If*

*End Sub*

*Public Overrides Sub CloseEditForm()*

*‘editwindow?.Close()*

*If editwindow IsNot Nothing Then*

*If Not editwindow.IsDisposed Then*

*editwindow.Close()*

*editwindow = Nothing*

*End If*

*End If*

*End Sub*


Further general model-specific functions, e.g., functions to manage and display the input and output connectors, a call function for the model image, and many more, are provided in the Appendix A.

#### 2.3.3. Implementation of Membrane Specific Functions

With the general model-specific functions created, a suitable environment to implement membrane-specific properties and functions is finally available. The following section shall give an insight into the approach to implementing functions that shall provide specific properties for the membrane model, e.g., the membrane area, calculation mode, flow mode, and functions to calculate stream-related properties.

A selection of the most relevant functions is provided and explained in this section.

Creating an entity of a stream is necessary at some points of the calculations. Therefore, a method to do so is introduced


*Public Function InitStream() As MaterialStream*

*Dim Value As MaterialStream*

*Value = New MaterialStream()*

*Me.FlowSheet.AddCompoundsToMaterialStream(Value)*

*Value.SetFlowsheet(FlowSheet)*

*Value.PropertyPackage = Me.PropertyPackage*

*Value.SpecType = StreamSpec.Temperature_and_Pressure*

*Value.ClearAllProps()*

*Return Value*

*End Function*


This function is applied to introduce an intermediate stream different from all input and output streams. It offers the opportunity to use the DWSIM integrated calculation routine to find the physical properties of mixed streams without changing anything of the actual present streams on the flowsheet.


*Public Function RefreshStream(Stream As MaterialStream, Temperature As Double, Pressure As Double, MolarFraction As Double(), MolarFlow As Double()) As MaterialStream*

*Stream.ClearAllProps()*

*Stream.SetMolarFlow(SumY(MolarFlow))*

*Stream.SetOverallComposition(MolarFlow.ToArray)*

*Stream.SetTemperature(Temperature)*

*Stream.SetPressure(Pressure)*

*Stream.SetFlashSpec(“PT”)*

*Me.PropertyPackage.CurrentMaterialStream = Stream*

*Stream.Calculate()*

*Stream.Validate()*

*Return Stream*

*End Function*


Especially the pervaporation process is highly dependent on the continuous calculation of the physical properties of the stream’s compounds. The actual calculation routine utilizes quantization steps within a single model. This causes a change in concentration when the stream is submitted to the following chamber. The function “RefreshStream” recalculates all stream-specific properties again. The partial pressure is a crucial variable for calculating the driving force. The partial pressure calculation follows two different routes depending on whether a gas permeation or pervaporation is simulated. The algorithm for gas permeation implements the law of Dalton.


*Public Function PartialPressure(MolFrac As Double(), Pressure As Double) As Double()*

*Dim Value(MolFrac.Length − 1) As Double*

*For I = 0 To MolFrac.Length − 1*

*Value(I) = Pressure * MolFrac(I)*

*Next*

*Return Value*

*End Function*


Even though the partial pressures of the vapor flow could be calculated, DWSIM’s internal functions were used to determine these values by getting the pressure of the feed stream.


*Public Function PartialPressure(Stream As MaterialStream) As Double()*

*‘get pressure and molarfrac from stream, apply Law of Dalton*

*Dim molarFraction As Double() = Stream.GetPhaseComposition(0)*

*Dim Pressure As Double = Stream.GetPressure()*

*Dim Value(molarFraction.Length − 1) As Double*

*For i = 0 To molarFraction.Length − 1*

*Value(i) = Pressure * molarFraction(i)*

*Next*

*Return Value*

*End Function*


Having the relevant partial pressures available finally allows calculating the composition y_i_ of the permeate on the retentate end.

Public Function yiRetentateMultids2(RetMolFrac As Double(), MolFracLastIt As Double(), ratio As Double(), Perm As Double()) As Double()


*Dim A(RetMolFrac.Length − 1), B(RetMolFrac.Length − 1), C(RetMolFrac.Length − 1), Value(RetMolFrac.Length − 1), ValueRaw(RetMolFrac.Length − 1) As Double*

*For I = 0 To RetMolFrac.Length − 1*

*A(I) = Perm(I) * RetMolFrac(I) * ratio(I) * (SumY(MolFracLastIt) - MolFracLastIt(I))*

*B(I) = 0*

*For J = 0 To RetMolFrac.Length − 1*

*If J = I Then*

*B(I) += 0*

*Else*

*B(I) += Perm(J) * ((RetMolFrac(J) * ratio(J)) − MolFracLastIt(J))*

*End If*

*Next*

*C(I) = Perm(I) * (SumY(MolFracLastIt) − MolFracLastIt(I))*

*ValueRaw(I) = A(I)/(B(I) + C(I))*

*Next*

*Dim SumRaw As Double = SumY(ValueRaw)*

*For I = 0 To RetMolFrac.Length − 1*

*Value(I) = ValueRaw(I)/SumRaw*

*Next*

*Return Value*

*End Function*


The value y_i_ provides the opportunity to calculate the stage-cut θ and consequently the retentate composition x_r_ and the corresponding partial- and vapor pressures.


*While (deviation >= 0.00001) And (count < 1000)*

*yi = yiRetentateMultids2(xr, yi, Ratio, Permeance)*

*pyi = PartialPressure(yi, pp)*

*For I = 0 To NumCompounds − 1*

*qi0(I) = qip(I)*

*Next*

*For I = 0 To NumCompounds − 1*

*sc(I) = (((Permeance(I) * (Area/Chambers))) * (((pxf(I) − pyp(I)) − (pxr(I) − pyi(I)))/(Log((pxf(I) − pyp(I))/(pxr(I) − pyi(I))))))/NEff(I)*

*Next*

*For I = 0 To NumCompounds − 1*

*qip(I) = sc(I) * NEff(I)*

*Next*

*qtp = SumY(qip)*

*For I = 0 To NumCompounds − 1*

*yp(I) = qip(I)/qtp*

*Next*

*pyp = PartialPressure(yp, pp)*

*qtr = 0*

*For I = 0 To NumCompounds − 1*

*qir(I) = NEff(I) − qip(I)*

*Next*

*qtr = SumY(qir)*

*For I = 0 To NumCompounds − 1*

*xr(I) = qir(I)/qtr*

*pxr(I) = Pf * xr(I)*

*Next*

*deviation = 0*

*For I = 0 To NumCompounds − 1*

*deviation = deviation + Abs(qip(I) − qi0(I))*

*Next*

*count = count + 1*

*End While*


This loop is the last instance of the actual calculations. It starts with an If- condition where the deviation of the stage-cut is compared. This error must fall below a tolerance of 10^−6^ to end the while loop, as long as the while-loop continues to run the calculations process with the determination of the permeate composition followed by its corresponding partial pressures. These calculations are followed by the determination of the stage-cut from $y_{i}$. $x_{r}$ and the remaining total flows are calculated from the stage cut.

The exportation of the corresponding file delivered the model into the “unitop” folder of DWSIM as a .dll file.

### 2.4. Test Cases

As a performance benchmark, this subsection shall introduce applied test cases in this work. The first test case is a polyetherimide/γ-alumina composite membrane characterized and described by Park et al. It was used to separate HAc, EtOH, EtOAc, and H_2_O. The initial model set-up was operated with a permeate pressure of 267 Pa. The fiber was further specified with an inner diameter of 7 mm and an arbitrarily large amount of fibers of 750,000 with 30 compartments. The length of the fibers was set to 1 m. An overview of the permeance of every compound and the feed, retentate, and permeate composition of the corresponding simulation of test case 1 is described in Section 4.1.

The model was also compared with other gas permeation and pervaporation solutions from the literature. These further test cases inherit both experimental results as well as simulated data. An overview of different investigated models is displayed in Table 1.

The second test case can be described as a hollow-fiber gas permeation membrane module that was deployed to separate a three-component mixture consisting of CO_2_, O_2_ and N_2_. Four different Modules were reported. However, only module 31 was compared in this study. The permeance of CO_2_ in the membrane was determined experimentally at 30 °C and 24 bar [37].

Test case three is a membrane model for multi-component gas permeation developed for Aspen Plus as a FORTRAN calculation routine. The model was compared to multiple models and experimental data. The test case was created with a simulation separating H_2_, N_2_, CH_4,_ and Ar with an upstream pressure of approximately 69 bar and a permeate pressure slightly above 11 bar. The permeances were taken from another publication specified in the work of Chowdhury et al. [38].

The final and fourth test case was derived from the work of Koch et al., who managed to separate EtOH, IPA, and ACE with a polymeric PERVAP^TM^ 1210 membrane with an active layer of polyvinyl alcohol receiving high purities of water in the permeate stream. The feed pressure was kept atmospheric, while the permeate pressure was adjusted to approximately 3000 Pa.

## 3. The Model

Staring DWSIM also initiates results in the usual loading screen, which is terminated with the initial menu, where a new file can be either set up or an existing one can be loaded. For this investigation, a new empty file was created. A random number of compounds can be chosen all of which shall have the same state. If the compounds are in a liquid form, the availability of interaction parameters must be considered. Nevertheless, the properties of test case 1 were chosen for this specific set-up. The membrane model can be found on the Separator/Tanks tab of the UO’s menu. The implementation of the flowsheet does not differ from the other UOs. The model offers the opportunity to attach two inlets, one of which is an energy inlet, and two outlet streams. The flow direction happens to be the configuration of a co-current model. The model is portrayed in Figure 2.

### Property Editor

The property’s editor was designed to improve the operability of the whole model itself. The editor of the model occurs just analogously to the other model’s editors by clicking on the model itself. Alongside the general information and the group of connections, there is also a box element with three tabs. Another box with only two tabs follows this box. The property table is portrayed in Figure 3.

The major calculation parameters are to be found on the first Table 2 drop-down menus with three entries each that are followed by the six fields with values that can be set here. The permeate pressure as well as three further fields with fiber-related properties are revealed on this part of the editor. Fiber is the primary area determining property. Further, the number of compartments is to be specified on the 5th field.

Toggling through the tabs by clicking delivers a list of compounds handled by the membrane and a table with the corresponding permeances.

The only function of the compounds tab is the opportunity to check specific compounds of the simulation. If a particular compound is checked, it will be considered in the separation process. Figure 4b is derived from the compounds checked in the tab before and only displayed the compounds and the lists which have been checked. The initial values for the permeances of the displayed compounds are 0 mol/m^2^sPa.

## 4. Results and Discussion

The following section shall give an insight into the behavior of the developed UOs in dependence on various parameters. Further, the accuracy and reliability of the results shall also be investigated.

### 4.1. Influence of Parameters

As mentioned in Section 1, the permeance of a compound through a specific membrane is a highly influential parameter for the flux of a compound through a membrane. Therefore, the behavior of the model, which uses the solution-diffusion model as a calculative background, is of interest. The permeance must be a highly influential parameter for the calculative background. The results of the test case one simulations are listed in Table 2.

**Table 2 membranes-12-01186-t002:** An overview of the permeance of every compound and the feed, retentate, and permeate composition of the corresponding simulation of test case 1. Permeance data obtained from [40].

Compound	Permeances [mol/m^2^sbar]	Feed mol%	Permeate mol%	Retentate mol%
HAc	0.000201	14	1.07	14.98
EtOH	0.000161	14	1.25	14.97
EtOAc	0.000125	36	3.01	38.51
H_2_O	0.00647	36	94.68	31.54

The permeance of H_2_O is seen to be at least ten times higher than the permeance of any other compound. Alongside this, it can also be observed that the composition of the permeate consists of almost 95% water, followed by EtOAc with 3%. HAc and EtOH share approximately 1% of the permeate composition. The composition of the retentate is quite similar to the feedstock. All compounds seem to be enriched with the exception of H_2_O, which dropped to a composition of 31.54%. The stage-cut was calculated to be 0.071.

#### 4.1.1. Area

Alongside the permeance of each compound, the available area of the membrane is decisive for the flux and composition of the product streams. Sensitivity analyses with the number of fibers as a variable were carried out to obtain an overview of the influence of the area on the simulation. The compositions of the permeate and retentate are displayed in dependence on the stage cut in Figure 5.

The initial behavior regarding the permeate reoccurs again, showing that the H_2_O content in the permeate is the highest at any stage-cut, followed by EtOAc with a very long distance. EtOH and HAc almost share the same compositions, close to 1% for all stage cuts < 0.3. While the behavior of the initial simulation in 3.2.1 was basically reproduced for the counter-current configuration, slight differences can be observed if the simulation was run in co-current mode. However, even though every compound permeates concentrations are still deficient compared to H_2_O, two compounds switch their positions in the rankings of the permeate molar fraction. EtOH surpasses HAc on the co-current simulations.

Further information about the model’s behavior is displayed by the decrease of H_2_O in the permeate and the corresponding increase of the other compounds. It can be observed that the trends shown are clearly non-linear in dependence on the stage cut. Since the permeate composition does not portray a complete picture of the system, the retentate composition is also displayed in Figure 5.

Two interesting trends are revealed in Figure 5. The trend for the H_2_O content in the retentate is driven down to a value very close to 0 at a stage cut of 0.4. The second information in this plot is revealed as the initial rise of other compounds concentrations up to a stage cut of approximately 0.4.

Another notable behavior is the fact that the simulations do not converge after a further increase in the number of fibers. This behavior may also be related to the low residual water content in the feed/retentate, resulting in a deficient driving force.

#### 4.1.2. Feed

The performance of the same membrane was also investigated numerically in dependence on the feed by increasing the EtOH content of the feed from 0.1 to approximately 0.7 wt%. A comparison of y_p_ and x_r_ for the counter-current and co-current is displayed for the permeate and the retentate streams in Figure 6.

The share of fast permeating H_2_O is the largest, starting with above 95 mol%. However, the rising share of EtOH in the feed leads to reduced water content in the permeate. While the composition of EtOAc and HAc stay almost constant in the permeate stream, a comparably strong rise of EtOH is revealed with rising x_F, EtOH_. Figure 6 also reveals a seemingly linear depletion of all compounds in the retentate stream with the exception of EtOH which is displayed with a strong rise to a similar scale as x_F, EtOH._ Further, also the stage cut is revealed to decline in both flow configurations with increasing x_F, EtOH_.

#### 4.1.3. Cells

Many models follow the approach to calculate the membrane as a single unit. Since the retentate and also the permeate composition is highly dependent on the change of the feed composition proceeding through the membrane, it can be assumed that a model could be tuned by a separation into multiple compartments. These compartments were realized as cells. The influence of the number of cells is displayed in Figure 7.

Specifying here what is visible in the images, the trends display a significant deviation for the compositions in the permeate for less than five cells. Whenever the amount of five cells is surpassed; the concentrations stay constant with comparably low deviations.

### 4.2. Comparisons

Three other test cases, cases 2, 3 and 4, were prepared and simulated. The test cases are covered and described in Section 2.4 and Table 1.

#### 4.2.1. Gas Permeation

Two test cases were carried out as gas permeation simulations. Test case 2 was simulated, resulting in a comparison between the experimental data obtained by Sada et al., and the DWSIM simulation is displayed in Figure 8.

As a first impression, it can be seen that the trends of the experimental data are in accordance with the trends of the simulations, while the actual CO_2_ concentrations also range in the same scale. The CO_2_ content of the permeate decreases with the rising stage cut. The deviation between the simulated data and experimental values is <0.84% for all simulations.

Test case three is another comparison for gas permeation. A comparison between the DWSIM model and the work of Chowdhury et al., is portrayed in Figure 9.

For practical reasons, the scale of the *y*-axis was cut into two segments giving better visibility of the plots. As indicated by the permeances of the available components, the composition of the permeate mainly consists of H_2_. It starts at approximately 97.5% for a stage cut of 0.3 and proceeds to decrease the more the flux increases. Correspondingly, the composition of the other compounds starts at just approximately 1% each and increases with the N_2_, CH_4_, and Ar in decreasing order. The most exciting information in Figure 9 is the seemingly good agreement between the data delivered by DWSIM and the simulated results of Chowdhury et al. [38].

#### 4.2.2. Pervaporation

The separation of liquid–liquid mixtures received less attention in simulation tools, as already mentioned at the beginning of the work. Yet, a test case for the pervaporation was set up for comparison purposes with this model with test case 4, which is based on the work of Koch et al. A comparison of the water composition of the permeate depending on the water fraction in the feed is displayed in Figure 10 [39].

Both curves show the same trend declining from considerably high purities of 99 wt% of water in the permeate. The experimental values’ slope is comparably low compared to the simulated. The maximum error between the experimental and the simulated results for w_p,H2O_ is still <1%.

## 5. Conclusions

This article has given a substantial description of the design and operation of a membrane model, which can simulate the separation of gaseous and liquid mixtures. The design enables comparably easy implementation of the model into DWSIM, including its property editor. However, due to the object-oriented design and general approach to the proceeding, the model may be integrated into every PS tool that can interact with custom solvers. In particular, this model based on open-source software is not just limited to prominent players in the field of the process engineering industry. Further, due to its convenient applicability, the model may also be considered suitable for didactic purposes, including subsequent development. The model’s performance reveals consistent and credible trends according to the significant influences of its primary mode.

Further, comparisons with experimental data unveil a comparably adequate predictive performance of the model for gas permeation operations. The accuracy of the pervaporation simulations is distinctly lower, which may be attributed to the more extensive influence of physical effects, such as the effects of friction and other impacts due to its more significant dependence on temperature. Yet, the major objective of a design of an integrated and modular membrane model with counter- and co-current flow configuration has been achieved. The basic algorithm of the code still allows enhancements and upgrades.

A proposal for such an upgrade would be the implementation of a fourth tab for the property editor, which shall inherit check-boxes for additional physical effects, e.g., the equation of Hagen–Poiseuille, an opportunity to provide the Reynolds number and the permeance as a function of temperature. Another perspective could be the implementation of estimates along the lines of an artificial neural network. Integrating more complex solvers or membrane types would also result in an increased simulation duration, which accurate predictions could mitigate.

## Figures and Tables

**Figure 1 membranes-12-01186-f001:**
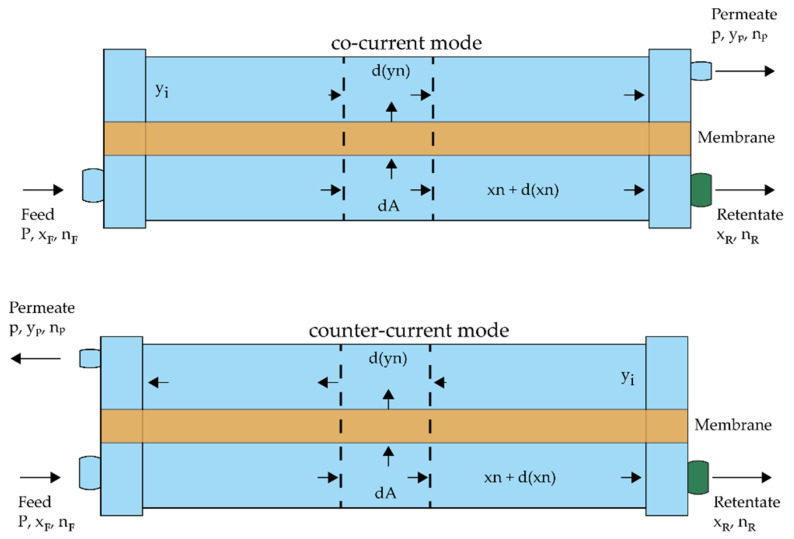
Illustrations of idealized co-current (top) and counter-current (bottom) flow patterns.

**Figure 2 membranes-12-01186-f002:**
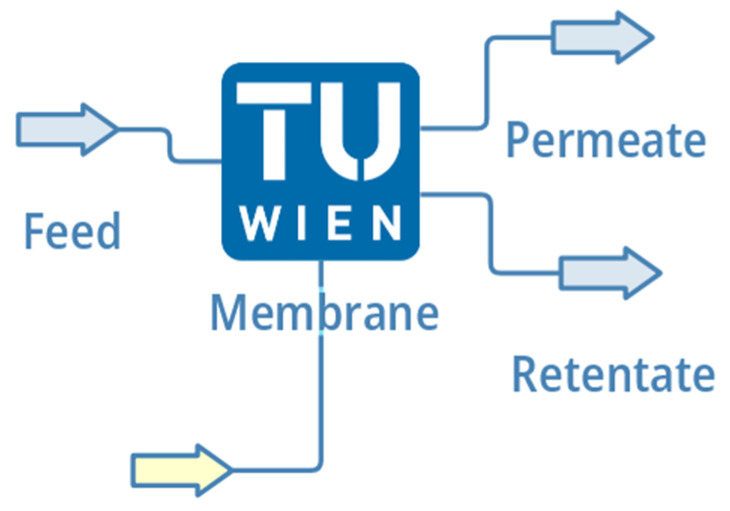
An actual impression of the unit operation on DWSIM’s flowsheet.

**Figure 3 membranes-12-01186-f003:**
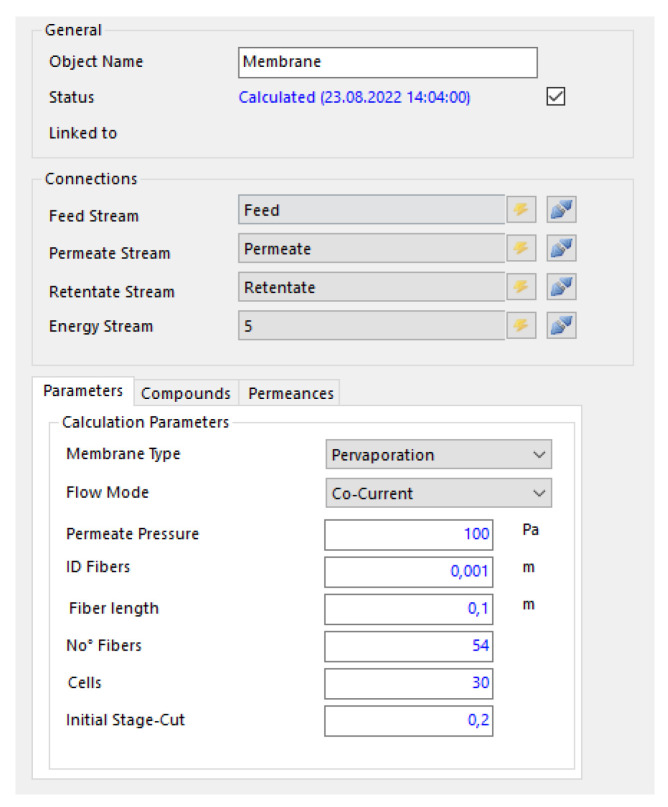
The property table with the first tab and its calculation parameters.

**Figure 4 membranes-12-01186-f004:**
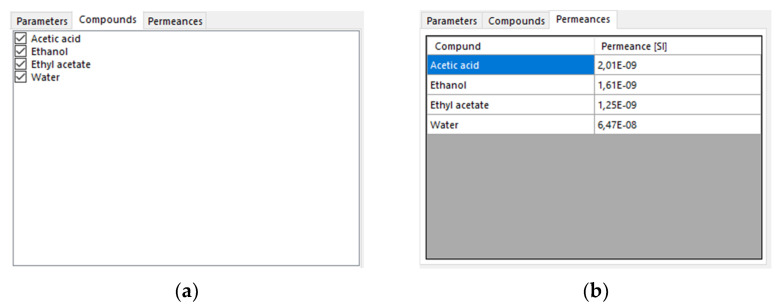
This figure portrays the compound (**a**) and the permeances (**b**) tabs of the membrane model.

**Figure 5 membranes-12-01186-f005:**
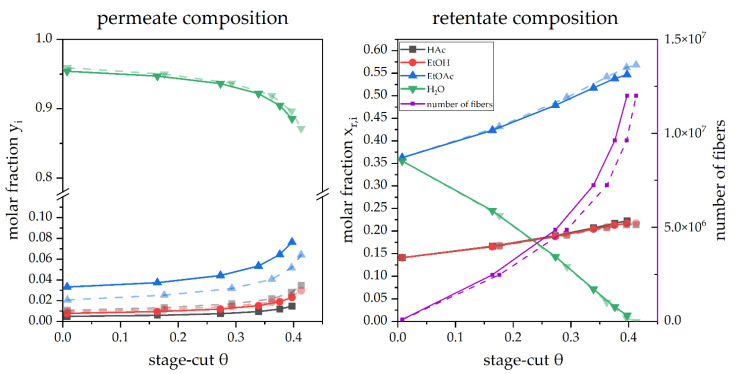
A simulated pervaporation operation in counter-current (
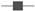
) and co-current (
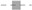
) mode in dependence on the stage cut θ.

**Figure 6 membranes-12-01186-f006:**
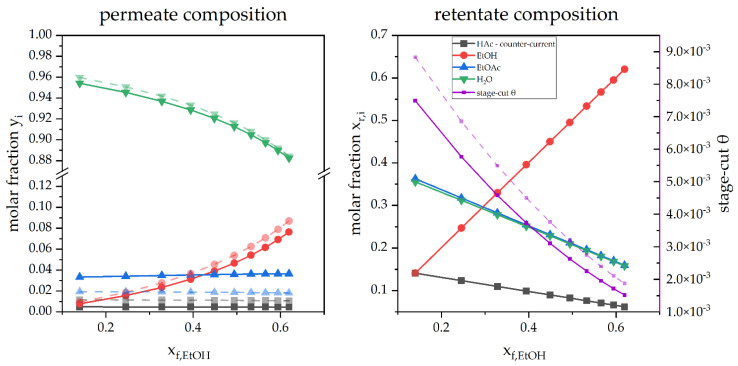
A simulated pervaporation operation in counter-current (
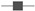
) and co-current (
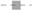
) mode depending on x_F, EtOH_.

**Figure 7 membranes-12-01186-f007:**
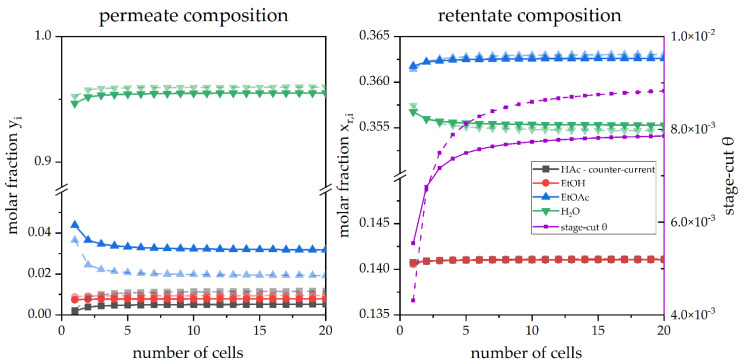
A simulated pervaporation operation in dependence on the number of compartments in counter-current (
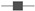
) and co-current (
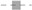
) mode.

**Figure 8 membranes-12-01186-f008:**
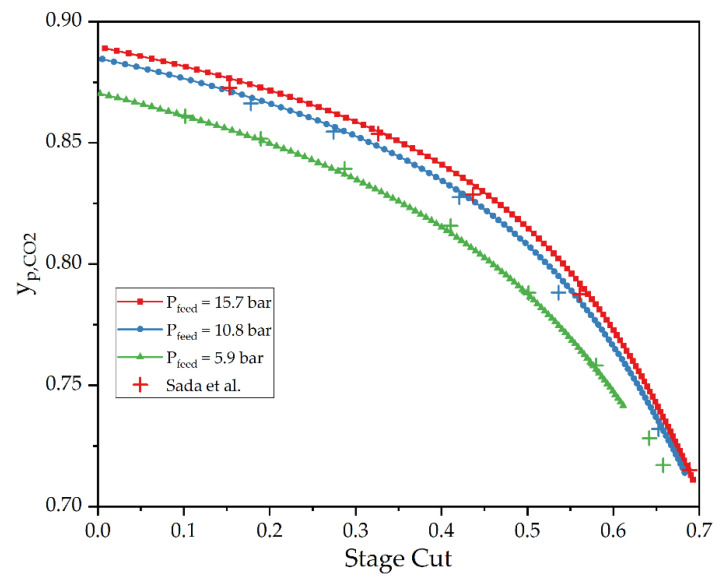
A comparison of an experimental gas permeation operation by Seda et al. [37]. (test case 2) and a DWSIM simulation in dependence of θ.

**Figure 9 membranes-12-01186-f009:**
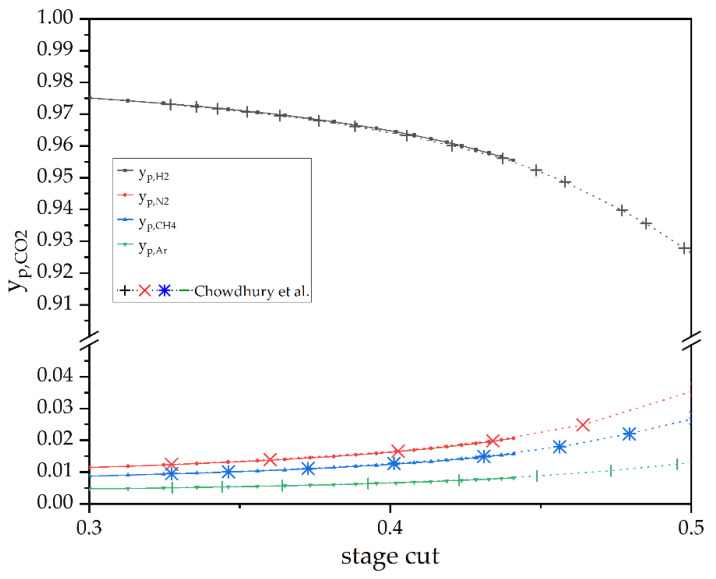
A comparison of an experimental gas permeation operation by Chowdhury et al. [38]. (test case 3) and a DWSIM simulation in dependence of θ.

**Figure 10 membranes-12-01186-f010:**
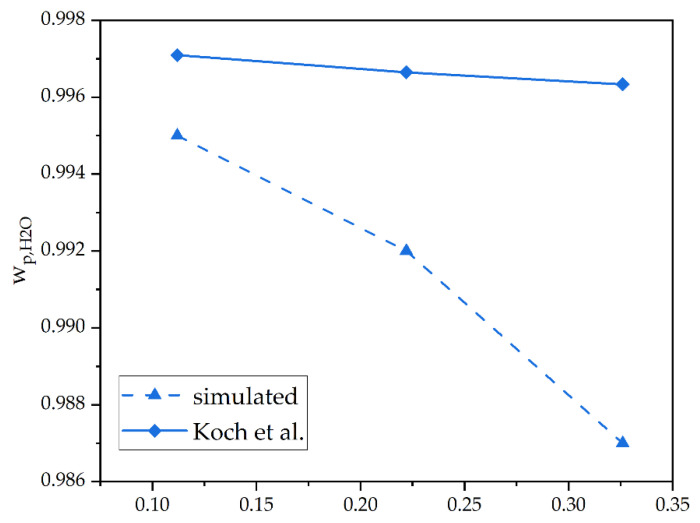
A comparison of an experimental pervaporation operation by Koch et al. [39] and a DWSIM simulation in dependence of w_F, H2O_.

**Table 1 membranes-12-01186-t001:** An overview of compared gas permeation and pervaporation systems.

	Sada et al. [37] Test Case 2	Chowdhury et al. [38] Test Case 3	Koch et al. [39] Test Case 4
membrane	asymmetric cellulose triacetate hollow fiber (Sample 31)	simulated asymmetric cellulose acetate hollow fiber	Hydrophilic polymeric PERVAP^TM^ 1210
type	gas permeation	gas permeation	pervaporation
flow configuration	counter-current	counter-current	co-current
inner diameter [µm]	0.000125	80	^1^ membrane area 159.4 cm^2^
length [cm]	63	15	
no. of fibers [-]	270	70	
Temperature [K]	303	298	~333 K
feed pressure [bar]	Varied between 15.7–5.9	69.64	
feed composition	50.0% CO_2_ 10.5% O_2_ 39.5% N_2_	51.78% H_2_ 24.69% N_2_ 19.57% CH_4_ 3.96% Ar	23.7 % ACE 65.1 % H_2_ 11.2 % IPA
permeate pressure [mbar]	1013.25	11230	
permeance [10^−10^ mol/s m^2^Pa]	CO_2_: 204.2 O_2_: 60.2 N_2_: 13.1	H_2_: 284 N_2_: 2.95 CH_4_: 2.84 Ar: 70	variable permeances

^1^ Values such as the inner diameter, number of fibers, and the fiber length are not provided by the source of the membrane.

## Data Availability

Not applicable.

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
