# Peer review of "Design of a Gas Permeation and Pervaporation Membrane Model for an Open Source Process Simulation Tool"

_membranes, 2022, doi:10.3390/membranes12121186_

Round 1

Reviewer 1 Report

Review membranes-1973850

In this manuscript, the authors propose a generalized numerical framework for solving cocurrent and counter- current gas permeation and pervaporation membrane models. The work is example of an intriguing area of research to increase feasibility of gas separation using advances in computational simulation.

The major concern arises due to lack of novelty since extensive works have been published in this area. Moreover, the model presented here uses rather simplified assumption, such as isobaric, isothermal and constant permeance operation, which has been proven to be a very idealized condition for simulation of membrane processes [1-3]. It is difficult to consider contribution of having an open-sourced membrane model as a novelty since the presentation is more like a technical report to guide users for coding in a commercial software, which has not developed by the authors themselves either. The results and discussion are mostly on validation and comparison with other published simulation data, which are pretty self intuitive considering that similar models have been used. The knowledge contribution and advancement for membrane community is lacking. The state of art review for mathematical solutions of membrane models is also very obsolete, which has been mostly dated more than a decade. Other issues include:

1)     In Table 1, the authors summarize specifications of the membrane in terms of inner diameter, length, no. of fibers, etc, which are not at all used/mentioned in the equations.

2)     In Figure 1, cocurrent flow should be shown to be consistent with authors description.

3)     Line 141-142, the abbreviation for feed pressure is missing. Variables used in equations (2) to (4), which are the main core of the methodology, are not properly defined, e.g., f, r, i, j, n again for total number of components, which is similar to molar flow?

4)     The whole manuscript should be also carefully checked to avoid editorial, language, grammar and syntax issues.

Hence, I cannot recommend publication of the manuscript in its present form mainly due to its lack of novelty and limited knowledge contribution to the membrane community.

References

[1] M. Safari, A. Ghanizadeh, M.M. Montazer-Rahmati, Optimization of membrane-based CO2-removal from natural gas using simple models considering both pressure and temperature effects, Int. J. Greenh. Gas Con., 3 (2009) 3-10.

[2] F. Ahmad, K.K. Lau, A.M. Shariff, Y.F. Yeong, Temperature and pressure dependence of membrane permeance and its effect on process economics of hollow fiber gas separation system, J. Membr. Sci., 430 (2013) 44-55.

[3] S. Lee, M. Binns, J.H. Lee, J.-H. Moon, J.-G. Yeo, Y.-K. Yeo, Y.M. Lee, J.-K. Kim, Membrane separation process for CO2 capture from mixed gases using TR and XTR hollow fiber membranes: Process modeling and experiments, J. Membr. Sci., 541 (2017) 224-234.

Author Response

Dear reviewer,

Best regards

Bahram Haddadi

Reviewer 2 Report

Based on my observation , the manuscript entitled "Design of a gas permeation and pervaporation membrane model for an open source process simulation tool" is interesting. However. some major concerns must be addressed to make it publishable. I recommend the acceptance of manuscript after major revision.

1) The authors must provide a detailed comparison between commonly-employed numerical/simulation-based models to highlight the advantages and disadvantages of each of them.

2) Section2. there is no chapter in an article. please revise it with "section"

3) the novelty of manuscript must be provided in the last paragraph of "Introduction".

4) English of manuscript needs revision.

5) Please improve the quality of figures.

6) More state-of-the-art articles about CFD-based articles in gas separation must be reviewed to enrich the manuscript.

7) I suggest the authors to provide a section about the role artificial intelligence technique in membrane processes.

Author Response

(The authors gave the same response as above.)

Round 2

Reviewer 1 Report

The authors have majorly addressed the comments. However, there are few minor corrections needed prior to acceptance:

Point 1: In Table 1, the authors summarize specifications of the membrane in terms of inner diameter,

length, no. of fibers, etc, which are not at all used/mentioned in the equations.

– This comment is still not addressed yet. The equation to calculate A from the inner diameter, fiber length, and the number of fibers need to be provided clearly in the manuscript. Else, it does not justify how Figure 8 is generated.

Point 2: There are still many obvious errors in the equations that need attention. Other than the listed below, please double check again for the entire manuscript.

In (3), the authors present an equation with ?i, x and y. Why some molar fraction has subscript and some not? Is ?not the same as y? Please check for the abbreviation also, why we do need to have a separate ?i and y.

In (6), please use ?F consistently for feed molar fraction and not a mixture of ?F and ?f.

Please check your abbreviation list again. Is ?p the permeate or retentate flow rate? Provide units for the abbreviation list.

Feed/ permeate weight fraction of component is defined in the abbreviation but not used/mentioned in the manuscript?

Author Response

(The authors gave the same response as above.)

Reviewer 2 Report

the authors have polished the manuscript well. therefore, it can be accepted in this Journal

Author Response

(The authors gave the same response as above.)
